# Constructive Optimization of Vulcanization Installations in Order to Improve the Performance of Conveyor Belts

**DOI:** 10.3390/ma12213607

**Published:** 2019-11-03

**Authors:** Dan Dobrota, Valentin Petrescu

**Affiliations:** Faculty of Engineering, Lucian Blaga University of Sibiu, Bulevardul Victoriei 10, Sibiu 550024, Romania; valentin.petrescu@ulbsibiu.ro

**Keywords:** conveyor belts, jointing, constructive optimization, finite element method, “bell”-type defect

## Abstract

Conveyor belts of special importance must have superior mechanical characteristics. The joining by vulcanization of the conveyor belts allows to obtain superior performances, but it has been found that at the vulcanizing joint of the conveyor belts, there is a “bell”-type defect. This type of defect can cause the quick removal of the conveyor belts from use; thus, within this paper, we realized the constructive optimization of vulcanization installations using the finite element method (FEM). Thus, the FEM analysis was performed for the installation used at the present time, which has four spacers for stiffening, moving to the next stage to a stiffening system with seven spacers and, in the last stage, it was proposed to use a stiffening plate. The joined conveyor belts were of type ST 2000, and by the constructive optimization of the vulcanizing press type DSLQ, the bell-type defect was greatly reduced. Also, an analysis of the effects that the constructive optimization of the vulcanization installation has on the resistance to extraction of the metallic insert from the rubber matrix of the costs determined by the proposed constructive modifications, has been performed.

## 1. Introduction

The conveyor belts used in different fields of industry can be made in the manufacturing process either at the required dimensions in exploitation or at other smaller dimensions, and this determines the use of vulcanization as a process for combining the conveyor belts so as to obtain the conveyor belts with the desired performances [1,2]. It is worth noting in this regard the need for joining of the conveyor belts that are used in the transport of materials over long distances. These strips have metallic inserts in structure and are used for the transport of coal and ores. Thus, the maximum lengths at which the strips with metal inserts are manufactured are made up to 210 m, due to difficulties that appear in the technological manufacturing process. Because this type of band has to transport materials at much greater distances than the manufacturing length, it is necessary to combine them by vulcanization [3].

Thus, the conveyor belts of special importance, which should have high mechanical properties, have different inserts (textile—Figure 1a, metal—Figure 1b), and their properties are based on rubber-insertion adhesions and are used for equipping high-capacity conveyors or those working in high demand conditions (high slope, high speed, reduced length with frequent alternations with straight paths and passes on drums) where very good breaking resistance, great flexibility and high reliability are required [4,5,6,7].

In order to increase the service life of the conveyor belts, it is necessary to combine them by vulcanization so that in the joining areas, the mechanical characteristics will be the same or very close to those of the rest of the belt and to obtain a homogeneity of the entire conveyor belt [8,9,10,11].

In general, the vulcanization of the various rubber products, but especially of the conveyor belts, generates lower mechanical characteristics in the joint area than the rest of the product and this causes a decrease in the life of the product [9]. In this sense, the aim is to find a technical solution that will allow the homogenization of the properties of the materials in the joint area with those of the material from the respective product.

The adjustment of the vulcanization time of the rubber products, in order to combine them, is made according to the thickness of the rubber in the area of their connection. In order to regulate the vulcanization temperature, the type of rubber used in the vulcanization process is taken into consideration, and the pressing pressure is determined according to the thickness of the rubber and its structure [10,11,12]. Any vulcanization process of a rubber product is currently carried out by regulating three technological parameters, namely:1)vulcanization time;2)vulcanization temperature;3)pressing pressure of the ends of the product.

The three parameters mentioned above may take different values depending on a certain type of rubber product, and if the adjustment of these parameters is not achieved at optimum values then the results obtained regarding the characteristics of the joint area are not appropriate. The inadequate results are mainly determined by the presence in the joint area of a high porosity of the joint material but also of an uneven thickness of the conveyor belt, and this causes a decrease of the mechanical characteristics of the joints [13,14,15].

The reduction of the porosity in the joint area can be obtained by the appropriate adjustment of the three technological parameters presented at optimal values [16]. In the process of jointing by vulcanization, the pressing pressure is one of the parameters of special importance. At present, different opinions have been issued regarding the influence of the pressing pressure, but in particular the research carried out refers to metallic materials and less to rubber materials or composite materials with rubber matrix [17]. An uneven distribution of the pressing pressure on the width of the conveyor belt can result in a non-corresponding quality for the realized joint [18].

A main cause that determines the removal of the conveyor belts is that in the joint area is not obtained an uniform thickness of the conveyor belt and thus the dimensional error of the type “bell” appears, meaning that the conveyor belt has in the joint area a much greater thickness in the middle compared to the outer parts. This fact is determined by the fact that the used vulcanization facility has an unoptimized metal structure that deforms very much during the vulcanization in the sense that it has a very large deformation in the middle [19].

In these conditions, during the research carried out within the work, the constructive optimization of the vulcanization installation was followed using the finite elements method so that the deformations suffered during this vulcanization were minimal and the appearance of the “bell-type” dimensional error was avoided. For the constructive optimization of the vulcanization joint installation, three constructive variants were considered, and after the researches, the constructive variant was established, which presents maximum rigidity and which allows the joint with the lowest dimensional deviation to be obtained. At the same time, an analysis of the effects of the constructive optimization of the vulcanization installation on the extraction resistance of the metallic insert from the rubber matrix of the costs determined by the proposed constructive modifications, was performed.

## 2. Materials and Methods

### 2.1. Materials Used in the Process of Jointing by Vulcanization of Conveyor Belts

For carrying out the experiments, there have been used general purpose conveyor belts from ATRBZ rubber (Targu Jiu, Gorj, Romania), Table 1, which has Ø5.85 mm diameter metal inserts in the structure. Such a conveyor belt has been chosen because it has to have a very good resistance to breaking, great flexibility and high reliability. Also, the conveyor belt under study is characterized by a width of 2000 mm and a thickness of 20 mm.

In order to achieve the vulcanization of the conveyor belts, the ATRBZ type rubber was used for the covering faces of the joined area - with a thickness of 5 mm; GDT type rubber, Table 2, with a thickness of 2 mm that was applied both on the inside faces of the joint area and on the cables left without rubber during the removal operations; the rubber of the type GDT with a thickness of 2 mm was located between the cables in the joint portion and between the ends of their remaining free places. There were also used vulcanization solutions GDT + 5% crosslinking agent of Desmodur R type produced by Covestro AG (Leverkusen, Germany) and trichlorethylene for pickling operations.

The cleaned metal insertions were placed, for protection, on an impermeable canvas on which two layers of GDT solution were mixed with cross-linking agent in the ratio of 1/4. Brushing should be done very carefully (the cables should be covered by the solution all around). The second layer of solution was applied only after complete drying of the first layer. The GDT solution applied to the cables has the same role as the core rubber of the belt—in order to ensure the necessary adhesion between the steel and rubber cables.

In order to determine the length of the joint, it was considered that it would be half the width of the band (at the two-stage joint adopted in the experimental research). Considering the type of conveyor belt chosen, namely ST 2000 (ARTEGO, Targu Jiu, Romania), it is necessary that the vulcanization joint to be made with the arrangement of the metal cables in two stages, Figure 2. In order to achieve a proper joining of the ends of the two conveyor belts, these were arranged respecting the following dimensions: l_v_ = 1350 mm—the length of the joint; l_q_ = 50 mm—the length of the deflection area of the metal cord; l_p_ = 20 mm—displacement of the metallic cord; l_st_ = 400 mm—the minimum step for the metal cord, l_s_ = 20 mm—distance between the ends of the metal cords.

### 2.2. Application of the Finite Element Method for Constructive Optimization of the Vulcanization Facilit

The research was carried out on a hot vulcanization press of the DSLQ conveyor belts, produced by the company Wagener Schwelm Corporation (Reisholzstraße, Hilden, Germany), which allows hot vulcanization of the rubber conveyor belts with metal cord, whith a width of up to 3000 mm. In the case of this type of vulcanization press, the pressure required in the vulcanization process is realized with the help of hydraulic cylinders. This type of joining press by vulcanization, of the conveyor belts, has the upper crossbars with hydraulic cylinders, and the lower crossbars and the heating plates are made of special high quality aluminum with high bending and traction resistance, to provide customers with a longer operation of the installations.The structural components of this machine are made largely of aluminum alloys. The main constructive characteristics of the DSLQ vulcanizing machine are:is compact, light in weight, reliable and easy to operate;the supply of the resistors is made of a triangular connection and the distribution;heating is uniform throughout the work surface;the press forces are realized with the help of a hydraulic installation which makes the press forces large and evenly distributed throughout the vulcanized surface;the thermal inertia is low due to the aluminum thermal plates;lower electricity consumption and high thermal efficiency through electricity supply, constant temperature and control of vulcanization times;the vulcanizing machine can be used for vulcanizing the belts of cloth, nylon or steel cables. It can also be used for the special vulcanization of anti-corrosion and heat-resistant belts. Can be used in metallurgy, mining, power plants, ports and places where there is no explosive or corrosive gas.

Regarding the main technical specifications of the vulcanization facility, these are the following:vulcanization pressure: 1.5 MPa. There is also the 1.8 MPa pressure variant;vulcanization temperature: 145 °C (adjustable);the temperature difference on the vulcanization plate +5 °C to −5 °C;heating period (from atmospheric temperature to vulcanization temperature): not more than 50 min;supply voltage: 380 V, 50 Hz (three-phase 4-wire) or 660 V;Output voltage from the electrical control box: 380 V, 50 Hz, with backup voltage: 220 V, 50 Hz;Output current of the control box: max. 30 A;temperature adjustment range: 0–300 °C;time adjustment range: 0–99 min;the difference between the electrothermal plate and the lower plate after pressing: not more than 0.5 mm.

The structure of the DSLQ type vulcanization press is shown in Figure 3.

As for the metallic structure of the vulcanization press, this is made up of the crossbars 1, the spacers 6 and the system for fixation of traverse package. These structural elements are made as follows: Al6061 aluminum alloy sleepers with the properties, according to [20], presented in Table 3; the spacer system for fixing of traverse package from C45 steel with the properties, according to [21], presented in Table 4.

During the vulcanization process, there are various mechanical demands in the components of the vulcanization facility. These loads are determined by the pressure introduced by the hydraulic installation used for vulcanization. The mechanical stresses of the components introduce tensions in them and they produce deformations. The presence of deformations influences the quality of the joints obtained by vulcanization and, in this sense, the deformations that have appeared in the vulcanization installation are worth noting. These deformations are determined by the bending stress and, thus, the deformation that appears is of the arrow type. The arrow type deformations consist of a deformation of the crossbars under the action of the loads produced by the hydraulic system because the crossbars are fixed at the ends, and in the middle of them there appears a maximum deformation of arrow type. The presence of this arrow-type deformation substantially determines the quality of the joint by vulcanization, showing a “bell”-type error, meaning that the conveyor belt has a greater thickness in the middle and smaller towards the edges. The presence of the bell-type error causes a decrease in the life of the conveyor belt due to the operating conditions, but also because the joint is not homogeneous.

In order to analyze the stresses and deformations (by the finite elements method) which appear in the constructive elements of the vulcanization facility at the beginning of the analysis, the restrictions and loads imposed on the vulcanization installation were established. Regarding the imposed restrictions, these are determined by the presence of the system for fixation of traverse package and thus the model analyzed was considered as double embedded. The metallic structure is requested for a series of loads determined by the action of the eight hydraulic pistons, distributed on the width of the vulcanization press.

In this case, due to the size of the surface of the conveyor belt and the vulcanization pressure, it was considered that the press requests the metallic structure of the 8-point vulcanization press (corresponding to the eight hydraulic cylinders), at a force of 78.500 N. In the calculations, only the loads produced by the hydraulic system were taken into account, because the crossbars fixing system with screws introduces insignificant tensions in the metallic structure of the vulcanization press in relation to it. Thus, in Figure 4 are presented the restrictions imposed on the vulcanization facility, and in Figure 5 is presented the way in which the loads act on the vulcanization installation during the vulcanization process.

In order to perform the analysis by the finite elements method (FEM) in the next step, it was necessary to discretize the constructive elements of the belt vulcanization press, and the way in which the discretization was done is presented in Figure 6. The discretization was achieved with the use of tetrahedron meshes and we considered the mesh with a dimension of 20 mm.

After realizing the discretization in the next stage, the tensions that appear in the components of the vulcanization installation and in the main way of distributing the tensions were analyzed, Figure 7.

The stresses that appear in the material of the construction elements of the vulcanization installation cause a series of deformations, and their size and distribution are shown in Figure 8.

From the analysis of the deformations that appear in the material of the components of the belt vulcanization press, Figure 6, it was observed that the maximum deformations appear in the central area of the crossbeams, and this explains the “bell”-type error that occurs when the conveyor belts are vulcanized. Also a relatively large deformation is obtained on the free ends of the crossbeams, and in order to avoid the occurrence of these deformations on the free ends of the crossbeams, the solution of disposing of stiffening plates on these ends was adopted, Figure 9.

The analysis by the finite elements method was performed for the current constructive variant of the vulcanization installation, which is characterized by the existence of four distances between the crossbeams. This system of stiffening of the vulcanization installation with spacers determines the appearance of the large deformations presented in Figure 7. In these conditions, in the next stage of constructive optimization of the vulcanization installation, it was aimed to increase its rigidity by intercalating between the four existing spacers, obtaining 3 more spacers, thus a more rigid structure is obtained, increasing the number of spacers between the crossbeams. The new metallic structure of the conveyor belt has been subjected to discretization by the finite element method, and the obtained structure is shown in Figure 10.

Following the discretization, an analysis of the stresses presented in Figure 11 was performed, and an analysis of the deformations is presented in Figure 12.

Following the results obtained for the analysis of stresses in Figure 1 and the deformations in Figure 12, we observed that by supplementing the number of distances, a reduction of the values of the tensions is obtained, but also a much better distribution of them in the sense that the beams have an approximately uniform distribution of the tensions in the central area which can lead to the reduction of the “bell”-type defect of the conveyor belts that occurs when the belts are vulcanized. Also, based on the data presented, Figure 12, there was observed a much lower value of the deformations in the active area of the vulcanization beams and a much better distribution of them if they are used for the stiffening of the sleepers a number of seven spacers. However, the introduction of seven spacers determines an increase of the deformations in the area where the holes are drilled to introduce additional spacers.

Considering the good results obtained in this stage of analysis of tensions and deformations in the next stage it was followed the adoption of a technical solution that will allow a further increase of the rigidity of the vulcanization installation, but at the same time the constructive solution adopted will be acceptable and from a technological point of view. Thus, at this stage the solution of replacing the seven spacers with a stiffening plate was adopted. The shape of this stiffening plate and the new constructive version of the belt vulcanization press is shown in Figure 13.

The new construction version of the vulcanization installation was subjected to analysis by the finite elements method in order to observe the new values of stresses and deformations that appear in the vulcanization installation. After realizing this constructive variant within the analysis by the finite elements method, there was performed the discretization of the vulcanization installation in Figure 14, the stress analysis in Figure 15 and the deformations analysis in Figure 16.

Following the analysis by means of the finite elements method (MEF) of the three constructive variants of the vulcanized press, used in the joining of the conveyor belts, the maximum values for von Mises stress of the deformations were obtained, Table 5.

## 3. Experimental Results

### 3.1. Technological Parameters Used at Joining Conveyor Belts by Vulcanization

In order to make the joints by vulcanizing the conveyor belts, the heating was done using vulcanization plates supplied with electricity. Also, the pressing force was made using hydraulic sleepers and initially a pressure of 80 bars was set. With the increase of the temperature of the heating plates the pressure of the vulcanization sleepers was increased, in this way:
Up to 50 °C80 barsUp to 80 °C125 barsUp to 100 °C250 barsUp to 125 °C380 barsUp to 145 °C400 bars

After the temperature of the heating plates exceeded the value of 100 °C, an increase of the pressure up to 380 bars was realized, parameter that was verified by the manometers mounted for this purpose on the installation.

With the further increase of the temperature of the vulcanization plates the oil pressure in the hydraulic installation of the vulcanization apparatus was increased up to 400 bars. The heating temperature was permanently monitored by thermostats and switching devices. In addition, a thermometer for temperature monitoring was introduced in each vulcanization plate. Also, it was taken into account that the thermometers in the vulcanization plate generally indicate a lower temperature by about 5–8 °C than the one existing on the heating surface, and this error will be taken into account in the vulcanization process. During the temperature monitoring, the vulcanization plates were disconnected or connected to the electricity supply so that their temperature uniformity could be achieved. Regarding the type of heating for vulcanization, it was calculated from the moment of reaching a temperature of the vulcanization plates of 125 °C. The vulcanization time for the conveyor belt joint was 70 min. After the expiration of the heating time, a stop and a cooling of the pressure joint were made up to about 60 °C, in order to avoid the possibility of the formation of any air bubbles inside the joint.

The experimental researches were performed using the three constructive variants of vulcanization installations, namely:—vulcanization installation in the classic construction version with four spacers for stiffening the crossbars (V1);—vulcanization installation in constructive version improved with seven spacers for stiffening the crossbars (V2);—vulcanization installation in constructive version when for a proper stiffening between the sleepers a stiffening plate is introduced (V3).

### 3.2. Analysis of the Bell-Type Defect that Occurs at the Joining by Vulcanizing of the Conveyor Belts

Respecting the mentioned parameters of the vulcanization process and using the three constructive variants of the vulcanization press, a number of three conveyor belts ST 2000, for each type of press, were joined by vulcanization. Following the vulcanization process, the deviation of the thickness of the conveyor belts in the joining area was measured. Thus, the thickness should be 20 mm and should be maintained at this value over the entire width of the conveyor belts. The deviation from the nominal thickness of the conveyor belts was measured with the aid of ultrasonic scanner for measuring material thickness NOVOTEST UT-1M (NOVOTEST, Novomoskovsk, Ukraine), that was set at zero at one end of the conveyor belt and moved over the entire width of the conveyor belt, and the measured values were recorded at distances from 50 to 50 mm. The data obtained were calculated as an average of the values of the deviations to thickness for the three conveyor belts joined with each of the three constructive versions of vulcanization presses. Regarding the maximum deviation to the thickness of the conveyor belts, this was 3.14 mm when using the V1 constructive variant of the vulcanization press, 1.05 mm when using the V2 constructive variant, 0.15 mm when using the V3 constructive variant. In these conditions, the change of the thickness of the conveyor belt was 15.7% for V1 variant, 5.25% in the case of V2 variant and 0.75% for V3 variant. The data obtained from the measurements were graphically processed, Figure 17.

From the analysis of the evolution of the dimensional deviations of the thickness of the conveyor belts, there was observed that the bell-type defect is very pronounced in the case in which the first constructive version of the vulcanization press (V1) is used and decreases if the V2 constructive variant is used, and if the V3 constructive variant is used, the bell-type defect is very small. The presence of the bell-type defect can also cause problems related to the cleaning of the conveyor belts, in the sense that the cleaning system achieves the complete removal of the mud only in the areas with high thickness of the conveyor belt. Also, an adjustment of the cleaning system, so that a full-width cleaning of the belt is carried out, causes an increased wear of the conveyor belt in the area of a greater thickness.

### 3.3. Determination of the Adhesion of the Rubber to the Metal Insert

The occurrence of the bell-type defect at the joined by vulcanization of the conveyor belts, can cause an inadequate vulcanization in the central area of the conveyor belt, and this causes the adhesion between the rubber and the metal insert to decrease. Thus, from the conveyor belts joined by vulcanization were taken samples from the central area of the band, but also from its peripheral area. For testing the adhesion of the rubber by the metal insert, samples of the form shown in Figure 18 were prepared.

In order to perform the adhesion testing of the rubber by the metal insert, a further preparation of the sample was required, in the sense that, the rubber layer in the clamping zone of sample in the test device was removed. Regarding the speed with which this test was done, it was set at a value of 100 mm/min. For the experimental researches, an equipment was used for testing metallic and non-metallic materials, controlled by the PC of type ATS 1630 CC (MATEST S.P.A., Treviolo, Italy). The determination of the adhesion between the metallic insert and rubber is based on establishing the specific extraction resistance of the metallic rubber insert:(1)$$R=\frac{E}{F},\;N/\mathrm{mm}$$
where: *F* is the extraction force, *L* = 100 mm—the length of the rubber matrix

The experimental results obtained for the joined conveyor belts with the help of the three constructive versions of vulcanization presses are presented in Table 6.

According to [22], the minimum extraction resistance, *R*, should be 85 N/mm. From the data obtained from experimental studies, it is observed that in the central area of the joint by vulcanization of the conveyor belt, a value of *R* is obtained below the minimum allowed limit of 85 N/mm, when using the constructive variants of vulcanization presses V1 and V2. Also, large differences were observed for *R* for the samples in the center of the edge of the conveyor belt for the constructive variants of vulcanization presses V1 and V2. If the V3 version of the vulcanization press was used the values of *R* is higher than the minimum values allowed and, at the same time, an uniformization of the values of the specific extraction resistance was observed starting from the center of the conveyor belt towards its edge. Thus, it is demonstrated that the presence of the bell-type defect also causes a decrease in the specific extraction resistance that may be a cause of premature removal of the conveyor belts.

Although the bell-type defect causes an increase in the thickness of the conveyor belt in the central area, this does not favor the increase of its lifetime by improving the abrasion resistance. This aspect can be explained by the fact that in the central area of the conveyor belt a superficial layer of rubber is removed very quickly, which has superior characteristics in terms of abrasion resistance, and further abrasive wear evolves much faster in the new rubber layers.

### 3.4. Cost Analysis Determined by the Constructive Optimization of the Vulcanized Press

In general, any constructive improvement made to an installation also determines its increases of manufacturing costs. Regarding the constructive variant V2, by increasing the number of spacers from three to seven, an increase of its manufacturing costs is obtained in relation to the variant V1. Thus, an assessment of the manufacturing costs of the vulcanization press is required for V2 and V3, respectively.

In order to carry out an analysis of the costs determined by the manufacture of the vulcanization presses in the V2 and V3 variants, the Ifind application provided by the company Sandvik Coromant (Sandviken, Gävleborgs Iän, Sweden) was used, which allowed the established of the manufacturing costs for a spacer and stiffening bar. This application allows us to obtain some information related to the cost of processing considering the material in the piece. For the manufacture of the spacers and for the stiffening bar, the C45 steel was used.

The data obtained using this application, for a spacer and stiffening bar, are presented in Table 7.

Based on the data presented in Table 6, it can be observed that the use of the seven spacers involves a cost of 221.9 Euro, which is lower compared to the cost of the stiffening bar, but this cost difference is much smaller compared to the subsequent economic effects produced by using the V3 version of the vulcanization press.

Based on those obtained in the experimental research stage, it can be concluded that the proposed objective has been achieved in the sense that the constructive optimization of the belt vulcanization press determines the obtaining of high precision joints with the elimination of the “bell”-type defect almost entirely. Also, it has been shown that eliminating the bell-type defect causes an increase in the specific extraction resistance, but also a decrease in the abrasion resistance of the conveyor belts. Thus, by this constructive improvement of the vulcanization presses, an increase in the operating performances of the conveyor belts is obtained.

## 4. Conclusions

Conveyor belts of special importance must have superior mechanical characteristics. The combination by vulcanization of the conveyor belts allows the obtaining superior performances, but it was found that, at the combining by vulcanization of the conveyor belts, a “bell”-type defect occurs. The bell-type defect is represented by the deviations presented by the conveyor belt in terms of thickness in the joint area. This type of defect causes the quick removal of the conveyor belts, due to the fact that these have a very high friction with the running lanes in the maximum thickness area. Preliminary research has shown that this “bell”-type error occurs due to the low rigidity of the belt vulcanization press.

Thus, for the constructive optimization of vulcanization installations, the finite element method (FEM) was use. Thus, the FEM analysis was performed for the installation used in the present time, which has four spacers for stiffening in the structure, passing to the next stage to a stiffening system with seven spacers, and, in the last stage, it was proposed to use a stiffening plate. The researches were carried out for the belt vulcanization press type DSLQ, and the conveyor belts that were joined were of the ST 2000 type.

When we used a rigging with seven spacers, there was a reduction of deformations in the area of active elements of the belt vulcanization press, but the deformations in the area where additional drilling operations were carried out, increased greatly. Thus, in order to avoid the need for further processing of the active parts of the belt vulcanization press, a new constructive variant was adopted, which involved the use of a stiffening plate. The obtained results showed that the use of a stiffening plate of the belt vulcanization press type DSLQ reduces the size of the “bell”-type defect, so that the thickness deviation of the conveyor belt decreased from 3.14 to 0.15 mm.

The new constructive version of the vulcanization installation can be used for the combination by vulcanization and to other types of conveyor belts, thus avoiding the occurrence of the “bell”-type error and substantially increasing the operating performance of the conveyor belts.

The constructive optimization of the vulcanizing press and the obtaining of the variant with the stiffening bar determines an increase and an uniformization of the values of the resistance to extraction, *R*, which allows an improvement of the performances of the conveyor belts in the sense of increasing their life. As for the manufacturing cost of the improved version of the vulcanized press (V3), it is higher than the other two analyzed variants, but this difference in cost is much smaller compared to the subsequent economic effects produced by using the V3 version of the press vulcanization.

## Figures and Tables

**Figure 1 materials-12-03607-f001:**
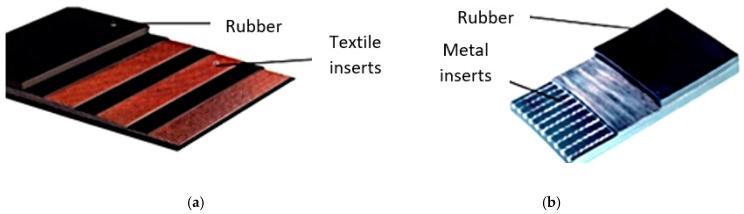
Structure of the conveyor belts. (**a**) with textile inserts; (**b**) with metal inserts.

**Figure 2 materials-12-03607-f002:**
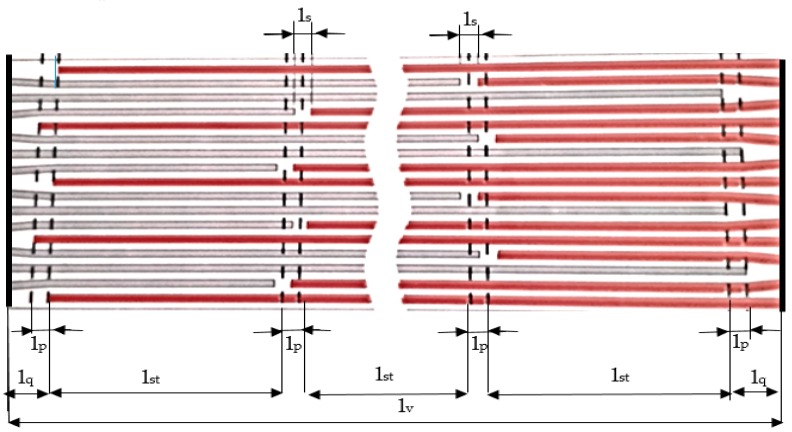
The way of arranging the cables at the vulcanization joint of the conveyor belt.

**Figure 3 materials-12-03607-f003:**
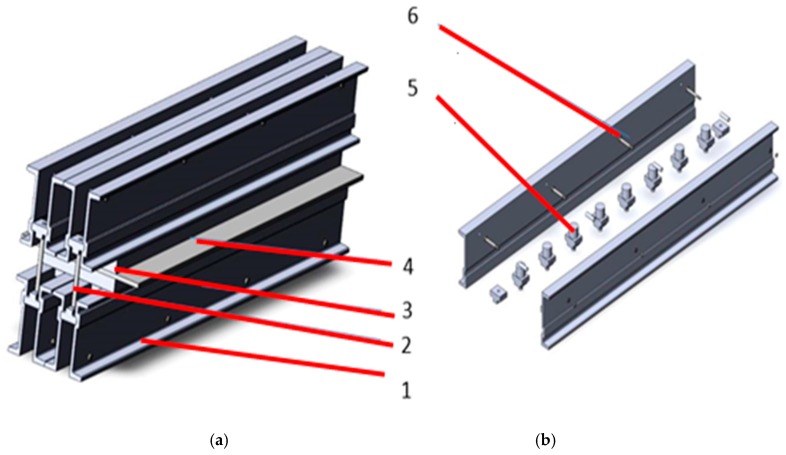
Structural presentation of the vulcanization facility. (**a**) general presentation of the belt vulcanization press; (**b**) presentation of constructive details of the belt vulcanization press. 1—traverse; 2—system for fixation of traverse package; 3—heating plate; 4—conveyor belt; 5—hydraulic pistons for maintaining the pressing pressure; 6—spacers.

**Figure 4 materials-12-03607-f004:**
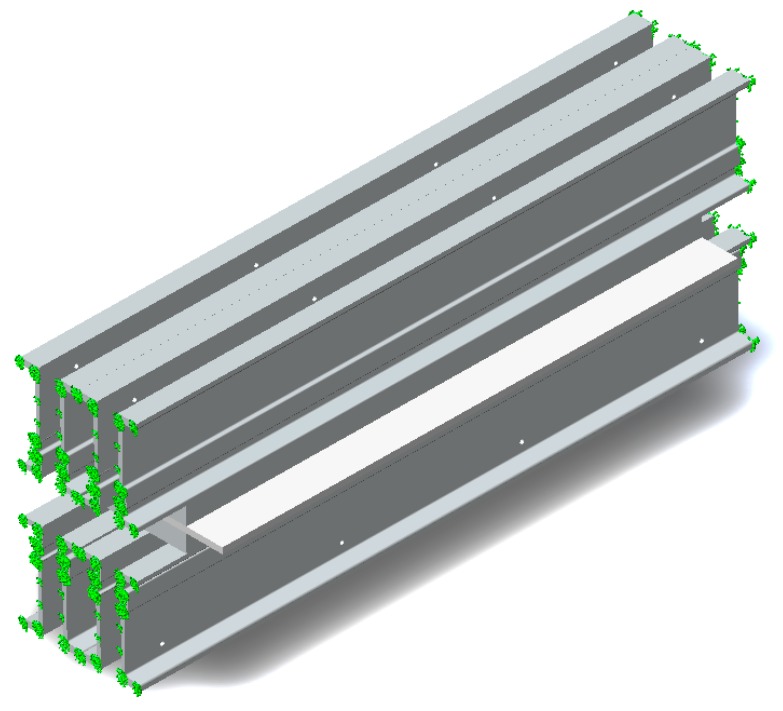
Restrictions imposed on the construction elements of the belt vulcanization press.

**Figure 5 materials-12-03607-f005:**
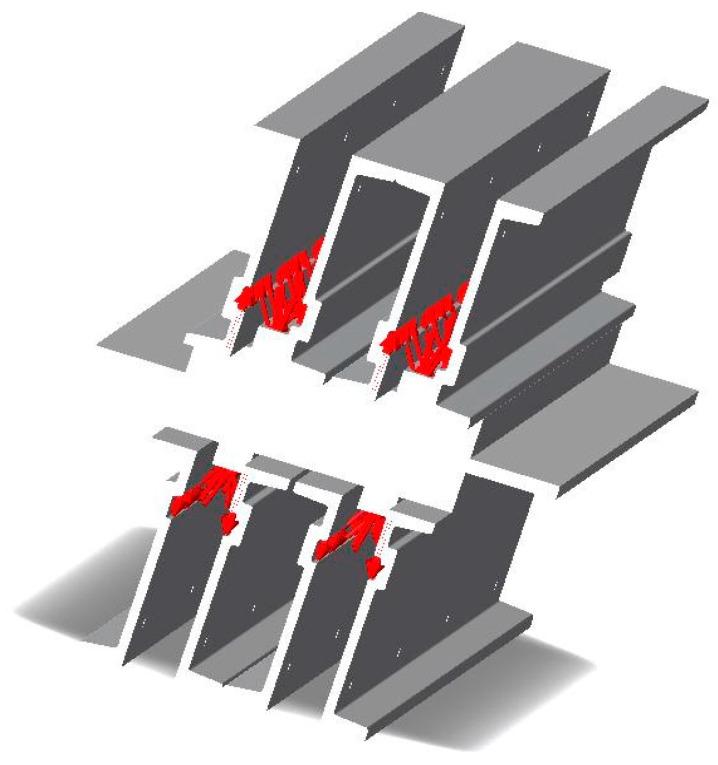
The loads to which are subjected the constructive elements of the belt vulcanization press.

**Figure 6 materials-12-03607-f006:**
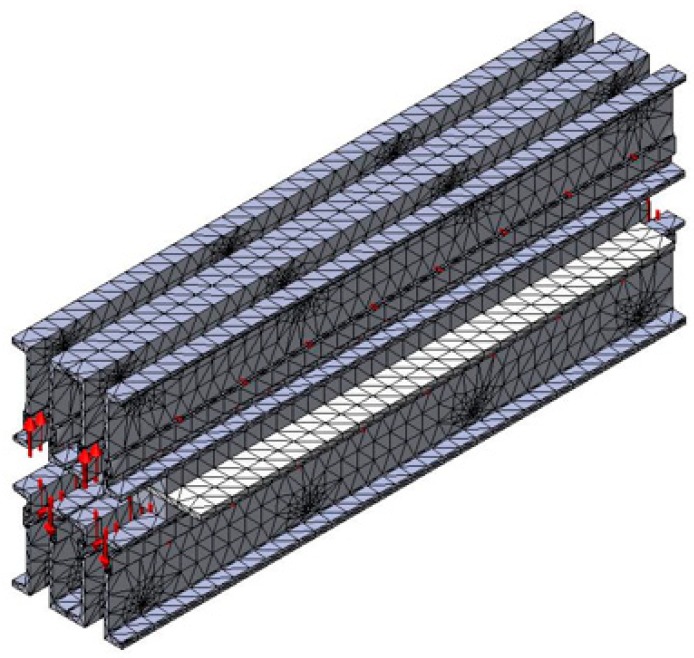
Discretion of the belt vulcanization press.

**Figure 7 materials-12-03607-f007:**
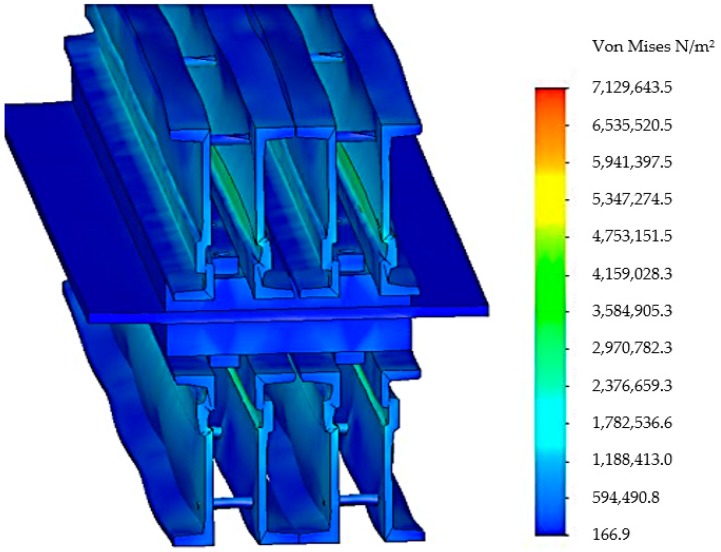
Distribution von Mises stress in the belt vulcanization press when using four spacers.

**Figure 8 materials-12-03607-f008:**
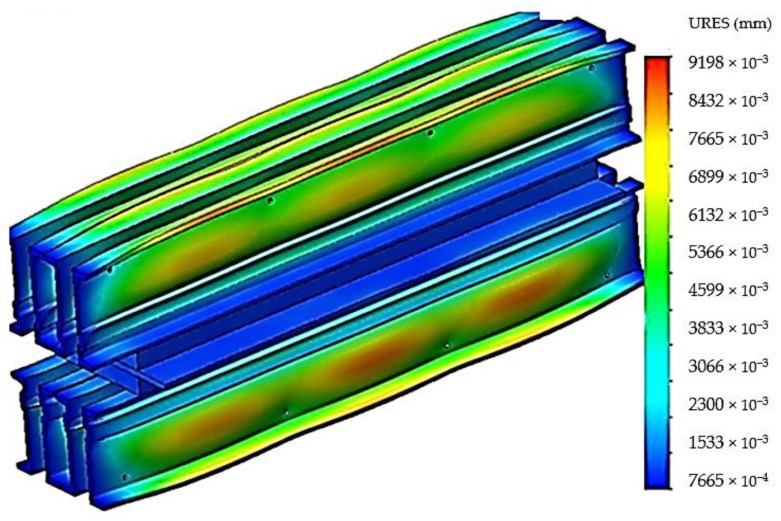
The deformations that appear in the constructive elements of the belt vulcanization press when using four spacers.

**Figure 9 materials-12-03607-f009:**
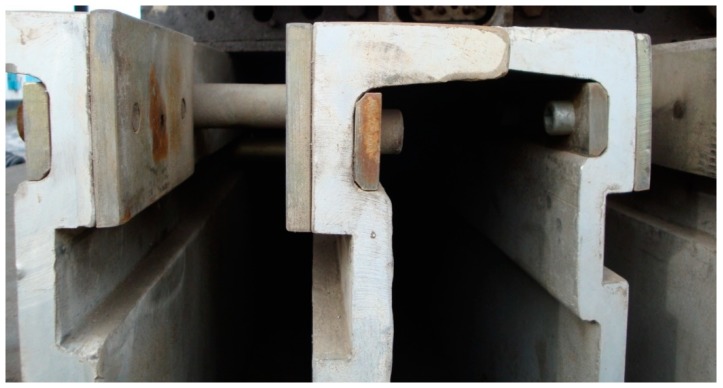
Arrangement of stiffening plates on the free ends of the crossbeams.

**Figure 10 materials-12-03607-f010:**
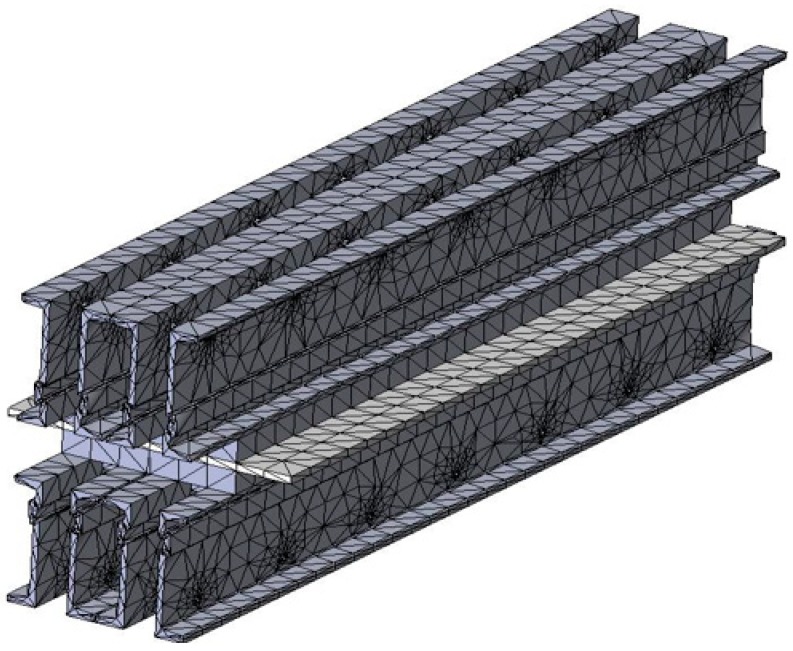
Discretion of the modified belt vulcanization press when seven spacers are used.

**Figure 11 materials-12-03607-f011:**
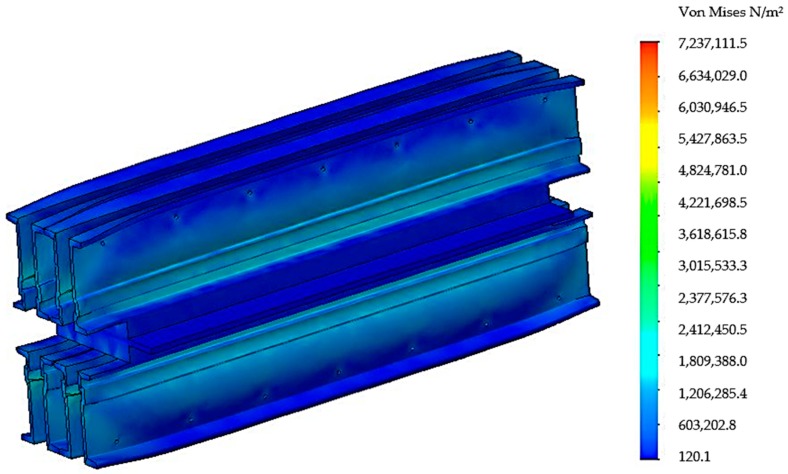
Distribution von Mises stress in the belt vulcanization press when using seven spacers.

**Figure 12 materials-12-03607-f012:**
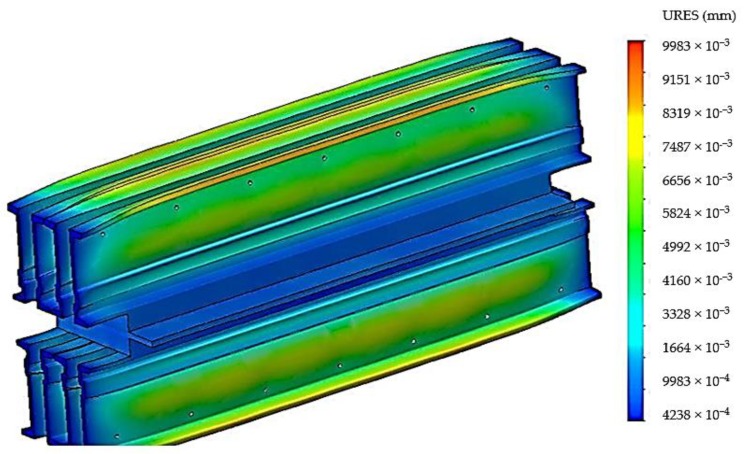
Deformation in the modified belt vulcanization press when seven spacers are used.

**Figure 13 materials-12-03607-f013:**
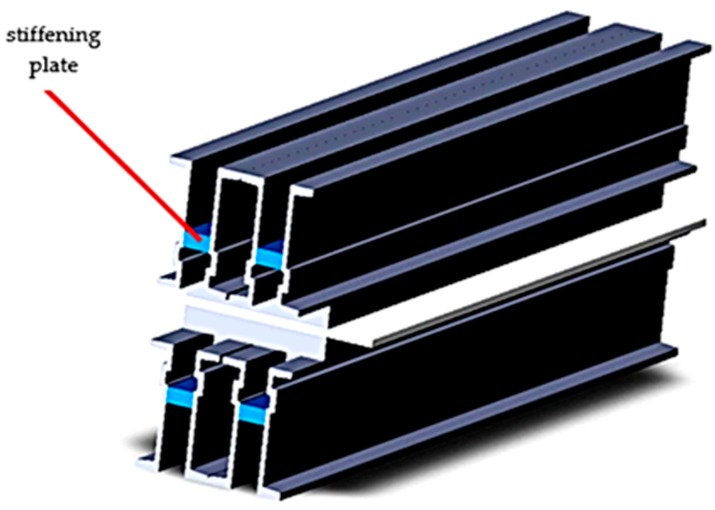
Belt vulcanization press with additional stiffening.

**Figure 14 materials-12-03607-f014:**
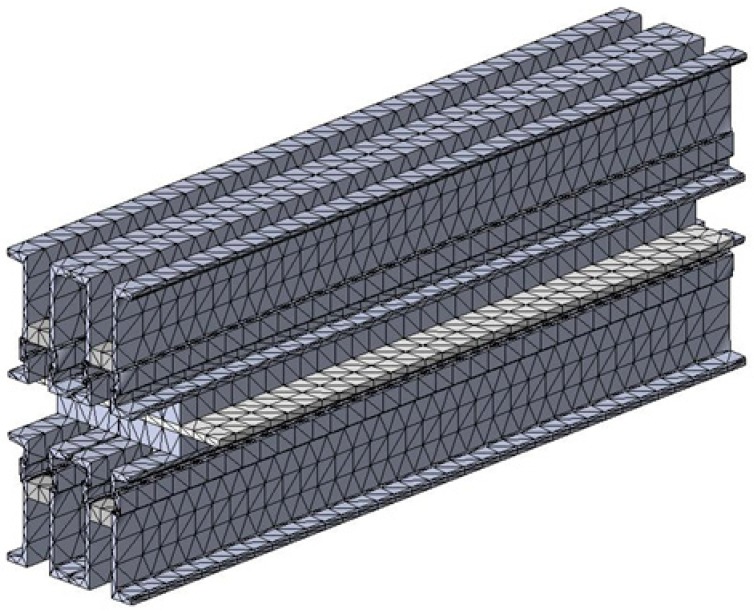
Discretion of the belt vulcanization press with the stiffening plate.

**Figure 15 materials-12-03607-f015:**
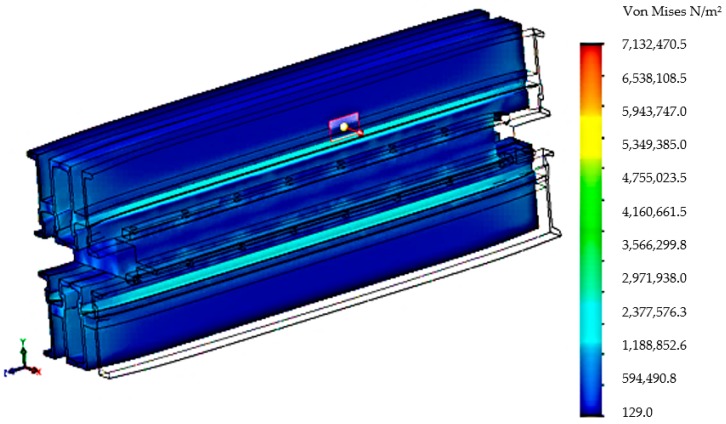
Distribution von Mises stress in the belt vulcanization press with the stiffening plate.

**Figure 16 materials-12-03607-f016:**
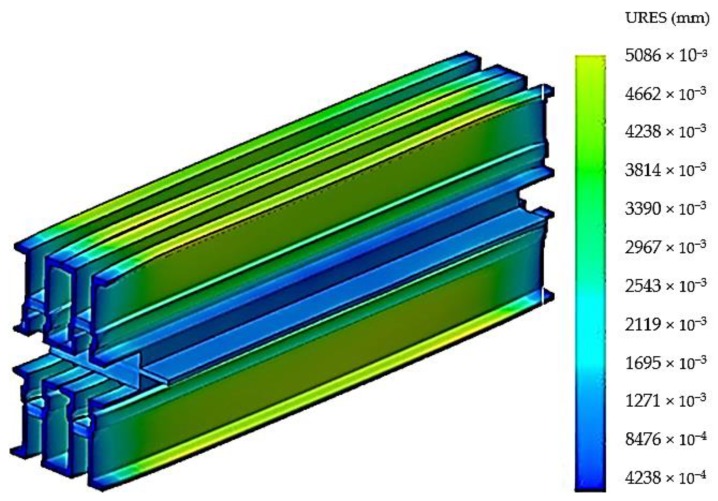
Deformation distribution in the belt vulcanization press with stiffening plate.

**Figure 17 materials-12-03607-f017:**
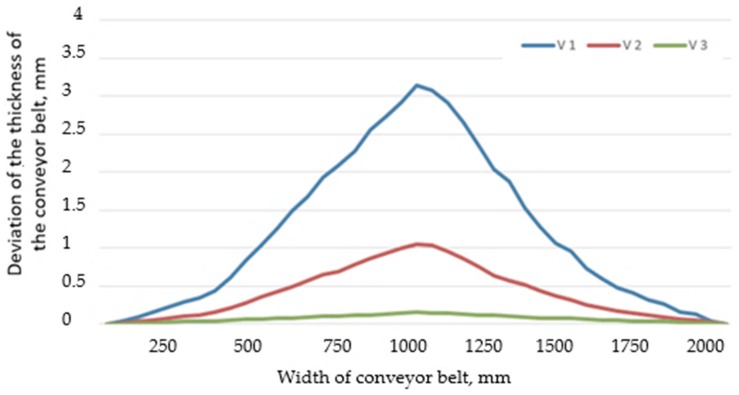
Evolution of the thickness on width deviation for the combined conveyor belts using the three constructive variants of the belt vulcanization press.

**Figure 18 materials-12-03607-f018:**
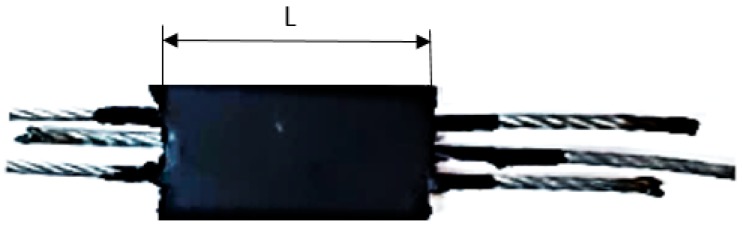
The shape of the sample used to determine the adhesion of the rubber to the metal insert.

**Table 1 materials-12-03607-t001:** The ATRBZ rubber structure.

Materials	phr
Natural rubber SIR-20	31
Styrene butadiene synthetic rubber SBR	32
Poly-butadiene synthetic rubber SKD ND	32
Reclaimed rubber	5
Naphtha	5
Carbon black HAF	5
Antioxidant 4010NA/LG(IPPD)	2
Stearin	2
Vulcanization accelerator DPG	2
Sulfur	3
Total	120

**Table 2 materials-12-03607-t002:** The GDT rubber structure.

Materials	phr
Natural rubber SIR-20	48
Styrene butadiene synthetic rubber SBR	31
Poly-butadiene synthetic rubber SKD ND	21
Naphtha	4
Carbon black HAF	4
Antioxidant 4010NA/LG(IPPD)	2
Stearin	2
Vulcanization accelerator DPG	2
Sulfur	3
Total	120

**Table 3 materials-12-03607-t003:** Aluminum alloy Al6061properties.

Properties	Value
Hardness, Brinell	95
Tensile strength	310 MPa
Yield Strength	276 MPa
Elongation at Break	17%
Modulus of Elasticity	69 GPa
Poisson’s Ratio	0.33
Fatigue Strength	96.5 MPa
Shear Modulus	26 GPa

**Table 4 materials-12-03607-t004:** C 45 steel properties.

Properties	Value
Hardness, Brinell	197
Tensile strength	600 MPa
Yield Strength	355 MPa
Elongation at Break	23%
Modulus of Elasticity	210 GPa
Poisson’s Ratio	0.3
Fatigue Strength	300 MPa
Shear Modulus	80 GPa

**Table 5 materials-12-03607-t005:** Mises stress of the deformations for the belt vulcanization press.

	Type of Vulcanization	Belt Vulcanization Press when Four Spacers Are Used	Belt Vulcanization Press when Seven Spacers Are Used	Belt Vulcanization Press with Stiffening Plate
Press Properties				
Von Mises stress, N/m^2^		7,129,643.5	7,237,111.5	7,132,470.5
Deformation, mm		9.198 × 10^−3^	9.983 × 10^−3^	5.086 × 10^−3^

**Table 6 materials-12-03607-t006:** The values of the specific resistance of extraction.

Specific Resistance of Extraction, R, N/mm					
Constructive variant, V1		Constructive variant, V2		Constructive variant, V3	
The sample in the center of the conveyor belt	The sample on the edge of the conveyor belt	The sample in the center of the conveyor belt	The sample on the edge of the conveyor belt	The sample in the center of the conveyor belt	The sample on the edge of the conveyor belt
73	97	83	98	95	98

**Table 7 materials-12-03607-t007:** The processing cost for a spacer and stiffening bar.

	Spacer	Stiffening Bar
Cost (Euro)	31.30	214.67

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
