# Peer review of "Constructive Optimization of Vulcanization Installations in Order to Improve the Performance of Conveyor Belts"

_materials, 2019, doi:10.3390/ma12213607_

Round 1

Reviewer 1 Report

In this manuscript, the authors discussed the technological aspects of optimization of vulcanziation system for conveyor belt. With the aid of FEM analysis, the authors optimized vulcanziation system to reduce “bell” type defect. It is recommended to publish this manuscript after the following minor revisions.

Please describe what type of rubber is ATRBZ rubber; The first sentence in “Conclusion” was repeated. Please correct.

Author Response

Dear reviewer,

We much appreciate your careful review. To improve the article, we have revised the article according to your suggestions. The changes and modifications in the manuscript have been highlighted.

Comment 1: Please describe what type of rubber is ATRBZ rubber. The first sentence in “Conclusion” was repeated. Please correct. 

Answer: See Table 1 respective Conclusions

               All changes are marked in red

Finally, we are very thankful to you for taking your valuable time to help us with this paper. Your insightful and constructive advice and recommendations are deeply appreciated.

Reviewer 2 Report

Description is illegible. The authors did not provide a diagram of the construction of a conveyor belt with a steel cables core and textile plies, but they write about „different inserts (textile, metal) in structure”, which in no way explains their construction to the readers. Not everyone knows how conveyor belts are built.

The paragraph regarding the construction of a closed belt loop from many belt sections connected together is also not clear. „The conveyor belts used in different fields of industry can be made in the manufacturing process either at the required dimensions in exploitation or at other smaller dimensions, and this determines the use of vulcanization”

The authors use the concept of „conveyor belts of special importance”, which suggests that they are not standard belts, but some special ones. They do not explain whether this “special importance” results from their design or the important role played by conveyor belts in the transport system.

It is not true that “A main cause that determines the removal of the conveyor belts is that in the joint area is not obtained an uniform thickness of the conveyor belt and thus the dimensional error of the type "bell" appears”. Main cause of joint removal is its elongation and threat that it will be broken during operation. The different thickness of the joint shows that the joint is not properly made but does not create a direct threat to its strength. Of course 16% difference in thicken of the whole belt and about 30% difference in thickness of belt’s cover can be a symptom that the pressure in area of so high cover/belt thickness deviation was not sufficient and condition of vulcanization process is worse than I other areas, but authors have not check the influence of such deviations on joint strength focusing on improvement of stiffness of construction of vulcanizing press. It is ok but importance of results of their work could be better justified by tests of such joints. Difference in cross section of joint area in my opinion is not enough. Some type of joints esp. multiplies belts have greater thickness deliberately. Abrasion is quicker in central area of the belt width so from this point of view greater thickness could be positive. However I think that greater thickness is a symptom that vulcanization of central area was poor and adhesion of rubber to steel cord is weaker. Therefore even thickness of joint on all its area proves that condition of vulcanization were even in every place.

Explanation that “This type of defect causes the rapid removal of the conveyor belts, due to the fact that these have a very high friction with the running lanes in the maximum thickness area” is not true. Conveyor belts are frequently covered by mud so they have increased thickness and cleaning devices are constructed in such a way that local increase of thickness is not a problem. The can adjust to uneven surfaces.

The authors use acronyms of different rubber types e.g. ATRBZ or GDT and DSQL  without their explanations.

They use term conveyor belt installation or vulcanizing machine instead of common term vulcanizing press. Not professional language is also used in other places what could lead to misinterpretation.

Fig.1 presents one of possible steel cord belt joints which is fact are standardized. The picture shows rather bad example of joint. We can not see proper steps and proper angle. One of possible reasons of weak performance of ST belts joints is their poor construction.

Fig.2 is not clear and is misleading. “1” is not a conveyor belt and it is difficult to recognize the ends of blue lines on the picture. There is no 4 in the description.

Authors use terms “vulcanization plant” and “vulcanization facility”. “Belt vulcanization press” is better and used in industry.

Results from Table 1 are presented on Fig.1 not 15. It would be a good idea to present also relative changes in joint thickness against belt thickness. .

I am not sure if Authors include in analysis the mechanism of putting even pressure inside heating plates by hydraulic pressure. It looks like only mechanical force from bolt tightening is included in analysis. We should observe to tensions and deformations: 1. after tightening bolts and 2. after putting hydraulic pressure to vulcanization plates.

The effort for finding improved construction of belt vulcanizing press is not justified by analyses of belt joint strength. Increased thickness is only potential cause of weak performance of belt joints. We do not know if increased thickness in central area of joint cross section is a consequence of lower pressure inside belt joint during vulcanization what can influence its performance (lower adhesion of cord rubber to steel wires). Authors have not modelled pressure inside belt. They have only modelled deformation in deformations in beams of vulcanization press. So only indirect cause was identified.

It is also difficult to base scientific conclusions based on only 1 sample of each joint.

There have been no description of belt thickness measurements methodology. We do not know if it has been only 1 point measurements or measurements are taken on all area? By which method? Using scanner or other device? 

Author Response

Dear reviewer,

We much appreciate your careful review. To improve the article, we have revised the article according to your suggestions. The changes and modifications in the manuscript have been highlighted.

Comment 1: Description is illegible. The authors did not provide a diagram of the construction of a conveyor belt with a steel cables core and textile plies, but they write about „different inserts (textile, metal) in structure”, which in no way explains their construction to the readers. Not everyone knows how conveyor belts are built.

Answer: Figure 1 and paragraph 1 of the introduction were added.

Comment 2: The paragraph regarding the construction of a closed belt loop from many belt sections connected together is also not clear. „The conveyor belts used in different fields of industry can be made in the manufacturing process either at the required dimensions in exploitation or at other smaller dimensions, and this determines the use of vulcanization”

.Answer: Paragraph 1 of the introduction has been completed.

Comment  3: The authors use the concept of „conveyor belts of special importance”, which suggests that they are not standard belts, but some special ones. They do not explain whether this “special importance” results from their design or the important role played by conveyor belts in the transport system.

Answer: Paragraphs 1 and 2 of the introduction have been completed.

Comment 4: It is not true that “A main cause that determines the removal of the conveyor belts is that in the joint area is not obtained an uniform thickness of the conveyor belt and thus the dimensional error of the type "bell" appears”. Main cause of joint removal is its elongation and threat that it will be broken during operation. The different thickness of the joint shows that the joint is not properly made but does not create a direct threat to its strength. Of course 16% difference in thicken of the whole belt and about 30% difference in thickness of belt’s cover can be a symptom that the pressure in area of so high cover/belt thickness deviation was not sufficient and condition of vulcanization process is worse than I other areas, but authors have not check the influence of such deviations on joint strength focusing on improvement of stiffness of construction of vulcanizing press. It is ok but importance of results of their work could be better justified by tests of such joints. Difference in cross section of joint area in my opinion is not enough. Some type of joints esp. multiplies belts have greater thickness deliberately. Abrasion is quicker in central area of the belt width so from this point of view greater thickness could be positive. However I think that greater thickness is a symptom that vulcanization of central area was poor and adhesion of rubber to steel cord is weaker. Therefore even thickness of joint on all its area proves that condition of vulcanization were even in every place.

Answer: Subsection 3.2 has been completed and 3.3 has been added

Comment 5: Explanation that “This type of defect causes the rapid removal of the conveyor belts, due to the fact that these have a very high friction with the running lanes in the maximum thickness area” is not true. Conveyor belts are frequently covered by mud so they have increased thickness and cleaning devices are constructed in such a way that local increase of thickness is not a problem. The can adjust to uneven surfaces.

Answer: Subsection 3.2 has been completed and 3.3 has been added

Comment 6: The authors use acronyms of different rubber types e.g. ATRBZ or GDT and DSQL without their explanations.

Answer: Table 1 and Table 2 were added and paragraph 2.2 was completed with paragraph 1

Comment 7: Fig.1 presents one of possible steel cord belt joints which is fact are standardized. The picture shows rather bad example of joint. We can not see proper steps and proper angle. One of possible reasons of weak performance of ST belts joints is their poor construction

Answer: Figure 1 has been modified. Now it is Figure 2

Comment 8: Fig.2 is not clear and is misleading. “1” is not a conveyor belt and it is difficult to recognize the ends of blue lines on the picture. There is no 4 in the description.

Answer: Figure 2 was modified. Now it is Figure 3.

Comment 9: Authors use terms “vulcanization plant” and “vulcanization facility”. “Belt vulcanization press” is better and used in industry.

Answer: This modification was made in all the paper..

Comment 10: Results from Table 1 are presented on Fig.1 not 15. It would be a good idea to present also relative changes in joint thickness against belt thickness. .

Answer: Table 1 was removed at the request of a reviewer. Subpoint 3.2 has been completed.

Comment 11: I am not sure if Authors include in analysis the mechanism of putting even pressure inside heating plates by hydraulic pressure. It looks like only mechanical force from bolt tightening is included in analysis. We should observe to tensions and deformations: 1. after tightening bolts and 2. After putting hydraulic pressure to vulcanization plates.

Answer: Section 2.2 has been completed (page 6).

Comment 12: The effort for finding improved construction of belt vulcanizing press is not justified by analyses of belt joint strength. Increased thickness is only potential cause of weak performance of belt joints. We do not know if increased thickness in central area of joint cross section is a consequence of lower pressure inside belt joint during vulcanization what can influence its performance (lower adhesion of cord rubber to steel wires). Authors have not modelled pressure inside belt. They have only modelled deformation in deformations in beams of vulcanization press. So only indirect cause was identified.

Answer: Subsection 3.2 has been completed and 3.3 has been added

Comment 13: It is also difficult to base scientific conclusions based on only 1 sample of each joint.

Answer: It was specified in point 3.2 that the results were obtained as an average of the measurements made for 3 samples.

Comment 14: There have been no description of belt thickness measurements methodology. We do not know if it has been only 1 point measurements or measurements are taken on all area? By which method? Using scanner or other device? 

Answer: Subsection 3.2, page 15, has been completed

All changes are marked in red.

Finally, we are very thankful to you for taking your valuable time to help us with this paper. Your insightful and constructive advice and recommendations are deeply appreciated.

Reviewer 3 Report

Dear authors,

Here are comments and remarks:

1. Is it possible to decode DSLQ abbreviation (page 3)?

2. Position 4 of Figure 2 is given in the description but the number "4" is missed.

3. Could you, please, give a scheme or a very brief explanation what the "arrow type" deformation is (page 4)?

4. Is it possible to specify the exact type of restriction you impose (Figure 3)? Can it be treated like "anchorage"?

5. It seems necessary to give values of loads to which equipment was subjected (Figure 4). Why exactly these areas (Figure 4) were chosen to apply loads?

6. Range of scales of Figure 6 and Figure 10 should be the same (if the interface of the software allows it doing) for easier comparing. Similar verification should be done for scales of Figure 7 and Figure 11.

7. Exact values of strain and stress range for three variants of simulations are better to give in the text or in a form of the table. It could clearly show which of the variants is most preferable in terms of lowering stresses and strains. 

8. Is it possible to clearly show where the stiffening plate is located on Figure 12?

9. it seems that units are missed for x-axis for "Width of conveyor belt".

10. Can all three variants be compared by costs and exploitation time? Using of 7 spacers instead of 4 and using of stiffening plate increases the cost of equipment significantly or not? 

Author Response

Dear reviewer,

We much appreciate your careful review. To improve the article, we have revised the article according to your suggestions. The changes and modifications in the manuscript have been highlighted.

Comment 1: Is it possible to decode DSLQ abbreviation (page 3)?

Answer: See paragraph 1, point 2.2, page 4.

Comment 2: Position 4 of Figure 2 is given in the description but the number "4" is missed.

.Answer: See the new Figure 3.

Comment 3: Could you, please, give a scheme or a very brief explanation what the "arrow type" deformation is (page 4)?

Answer: See paragraph 1, page 6.

Comment 4:  Is it possible to specify the exact type of restriction you impose (Figure 3)? Can it be treated like "anchorage"?

Answer: See completions in page 6, See paragraph 2.

Comment 5: It seems necessary to give values of loads to which equipment was subjected (Figure 4). Why exactly these areas (Figure 4) were chosen to apply loads?

Answer: See completions in page 6, last paragraph.

Comment 6: Range of scales of Figure 6 and Figure 10 should be the same (if the interface of the software allows it doing) for easier comparing. Similar verification should be done for scales of Figure 7 and Figure 11.

Answer: The scale for the indicated figures was restored

Comment 7: Exact values of strain and stress range for three variants of simulations are better to give in the text or in a form of the table. It could clearly show which of the variants is most preferable in terms of lowering stresses and strains. 

Answer: Table 5 was added

Comment 8: Is it possible to clearly show where the stiffening plate is located on Figure 12?

Answer: See now Figure 13

Comment 9:  it seems that units are missed for x-axis for "Width of conveyor belt".

Answer: See now Figure 17

Comment 10: Can all three variants be compared by costs and exploitation time? Using of 7 spacers instead of 4 and using of stiffening plate increases the cost of equipment significantly or not? 

Answer: The subpoint 3.4 was added.

             All changes are marked in red.

Finally, we are very thankful to you for taking your valuable time to help us with this paper. Your insightful and constructive advice and recommendations are deeply appreciated.

Reviewer 4 Report

Comments on “Constructive Optimization of Vulcanization Installations in Order to Improve the Performance of Conveyor Belts” by Dobrota Dan and Petrescu Valentin

In this work, the authors have performed both finite element simulations and experiments to study how to optimize the vulcanization installations to improve the performance of the convey belts. The combination by vulcanization of the conveyor belts allows the obtaining superior performances, but it was found that, at the combining by vulcanization of the conveyor belts, a "bell" type defect occurs. The bell type defect is represented by the deviations presented by the conveyor belt in terms of thickness in the joint area. Thus, for the constructive optimization of vulcanization installations, the finite element method (FEM) was use. Thus, the FEM analysis was performed for the installation used in the present time, which has 4 spacers for stiffening in the structure, passing to the next stage to a stiffening system with 7 spacers, and, in the last stage, it was proposed to use a stiffening plate. The researches were carried out for a DSLQ vulcanizing machine, and the conveyor belts that were combined were of the ST 2000 type. When was used a rigging with 7 spacers, there was a reduction of deformations in the area of active elements of the vulcanization plant, but the deformations in the area where additional drilling operations were carried out, increased greatly. Thus, in order to avoid the need for further processing of the active parts of the vulcanization plant, a new constructive variant was adopted, which involved the use of a stiffening plate. The obtained results showed that the use of a stiffening plate of the DSLQ installation substantially reduces the size of the "bell" type defect, so that the thickness deviation of the conveyor belt decreased from 3.14 mm to 0.15 mm. Over all, the whole manuscript reads like a lab report without enough details and scientific insights. It should be completely re-written. The authors should consider the following revisions:

Section 2.1 talks about the materials. But, none of these material properties are given, and how they are made. Section 2.2 is just a list of DSLQ machine. It is not clear what are the model set up, element type, material properties, boundary conditions, loadings, etc used in the FEM. For FEM, mesh convergence should be given and the dimension of these models should be given. Table 1 is not necessary and nobody can read it. A plot of these results should be enough.  

Author Response

Dear reviewer,

We much appreciate your careful review. To improve the article, we have revised the article according to your suggestions. The changes and modifications in the manuscript have been highlighted.

Comment 1: Section 2.1 talks about the materials. But, none of these material properties are given, and how they are made. Section 2.2 is just a list of DSLQ machine. It is not clear what are the model set up, element type, material properties, boundary conditions, loadings, etc used in the FEM. For FEM, mesh convergence should be given and the dimension of these models should be given. Table 1 is not necessary and nobody can read it. A plot of these results should be enough.   

Answer: Sections 2.1 and 2.2 were rewritten, but other information was added in the paper.

             All changes are marked in red.

Finally, we are very thankful to you for taking your valuable time to help us with this paper. Your insightful and constructive advice and recommendations are deeply appreciated.

Round 2

Reviewer 3 Report

All the required modifications were done by the authors.

Reviewer 4 Report

All the concerns have been addressed. The reviewer would like to suggest publication of this work.